# Insights into the Response in Digestive Gland of *Mytilus coruscus* under Heat Stress Using TMT-Based Proteomics

**DOI:** 10.3390/ani13142248

**Published:** 2023-07-09

**Authors:** Lezhong Xu, Yuxia Wang, Shuangrui Lin, Hongfei Li, Pengzhi Qi, Isabella Buttino, Weifeng Wang, Baoying Guo

**Affiliations:** 1National Engineering Research Center for Marine Aquaculture, Marine Science and Technology College, Zhejiang Ocean University, Zhoushan 316022, China; 2Italian Institute for Environmental Protection and Research ISPRA, Via del Cedro n.38, 57122 Livorno, Italy

**Keywords:** *Mytilus coruscus*, TMT-based proteomic, heat stress, digestion, metabolism

## Abstract

**Simple Summary:**

High-temperature stimulation can lead to severe stress response and affect the normal physiological function of animals. Generally, animals respond to the stimulation of extreme environments by regulating their physiological functions, including energy metabolism, biofactor synthesis, and degradation. The results of this study suggest that marine animals may respond to heat stress by regulating oxidative stress-related enzymes and basic metabolic levels.

**Abstract:**

Ocean warming can cause injury and death in mussels and is believed to be one of the main reasons for extensive die-offs of mussel populations worldwide. However, the biological processes by which mussels respond to heat stress are still unclear. In this study, we conducted an analysis of enzyme activity and TMT-labelled based proteomic in the digestive gland tissue of *Mytilus coruscus* after exposure to high temperatures. Our results showed that the activities of superoxide dismutase, acid phosphatase, lactate dehydrogenase, and cellular content of lysozyme were significantly changed in response to heat stress. Furthermore, many differentially expressed proteins involved in nutrient digestion and absorption, p53, MAPK, apoptosis, and energy metabolism were activated post-heat stress. These results suggest that *M. coruscus* can respond to heat stress through the antioxidant system, the immune system, and anaerobic respiration. Additionally, *M. coruscus* may use fat, leucine, and isoleucine to meet energy requirements under high temperature stress via the TCA cycle pathway. These findings provide a useful reference for further exploration of the response mechanism to heat stress in marine mollusks.

## 1. Introduction

Temperature is a crucial environmental factor that affects the survival of marine life. A suitable ambient temperature is beneficial to the growth, development, and reproduction of marine life [1,2,3]. However, if the temperature exceeds a certain range, it will have various negative effects on organisms, including decreased protein levels, decreased fertility, and inhibited immune systems [4,5]. With rising global temperatures [6], marine life, especially bivalves, are experiencing stress and death. In the warmest months of the southern summer, aquacultured New Zealand *Perna canaliculus* have suffered from mass die-offs [7]. In addition, due to the effects of high temperatures, mass mortality of oyster and the extensive infection of *M. edulis* with *Marteilia pararefringens* are common phenomena in summer [8,9].

*M. coruscus* is an economically important aquatic shellfish that is widely distributed in the coastal waters of the Yellow Sea and the East China Sea, especially in the coastal areas of Zhejiang [10,11]. In farmed *Mytilus,* compared to another aquatic shellfish, *M. galloprovincialis, M. coruscus* has the higher protein, crude fat, DHA, and trace element content, as well as a higher economic value [12]. With increasing temperatures, the yield and performance of *M. coruscus*’ byssus weaken, leading to shedding [13]. Additionally, the absorption and excretion rates of *M. coruscus* are reduced as the temperature increases [14]. Moreover, high temperatures reduce the growth and survival rate of *M. coruscus*’ larvae [15]. Therefore, the rise in temperature is detrimental to the survival of *M. coruscus*. Currently, studies on the effects of high temperature on *M. coruscus* mainly focus on physiological functions and specific functional genes [16], but the literature currently lacks studies on its heat resistance mechanism. Recently, we investigated the changes in transcription levels of mussels and speculated based on RNA-sequence analysis that mussels may respond to high temperature through ubiquitination and lysosome pathways [17]. However, it is well known that regulation of mRNA levels does not exactly correspond to regulation of protein levels, and changes in protein levels can reveal the true molecular regulatory process in the cell.

In aquatic animals, the key organ of metabolism is the digestive gland, which can respond to the stress of biological and abiotic factors through a series of physiological reactions [18,19,20,21,22]. To explore the regulation mechanism of *M. coruscus* in response to high-temperature stimulation, the enzyme activities related to immune stress and proteomic profiling of digestive gland were examined in *M. coruscus* growing at 18 °C, 26 °C, and 33 °C. Our results provide insights into the relative importance of these proteins and related biological pathways, which provides direction for research into the heat tolerance mechanisms of *M. coruscus*.

## 2. Materials and Methods

### 2.1. Animal Culture and Heat Treatments

*M. coruscus* specimens were sourced from an aquatic product market located in Zhoushan City, Zhejiang Province, China. The specimens were acclimated to the natural environment for seven days in an aerated water tank with a pH of 7.5–8.0, salinity of 30 ‰, and temperature of 18 °C. Three temperature groups of 18 °C, 26 °C, and 33 °C were set in this study as described by previous study [17]. After 24 h of exposure to different temperatures, the digestive glands of *M. coruscus* from each temperature group were collected in triplicate, and their proteins were immediately extracted.

### 2.2. Determination of Biological Enzymes

Each digestive gland sample was mixed with physiological saline at a ratio of 1:9. The resulting mixture was centrifuged at 465× *g* for 10 min at 4 °C. The supernatant obtained after centrifugation was stored at −20 °C for subsequent experimental analysis. All procedures were conducted on ice.

The activities of SOD, ACP, and LDH, as well as LZM content [23,24], were determined in the supernatant using assay kits according to the reagent guidelines. All the reagents used were from Nanjing Jiancheng Bioengineering Institute, Nanjing, China.

### 2.3. Labeling and Sequencing of Proteins

Each experimental group was composed of three replicates, with each replicate consisting of three mussels. The digestive gland tissue was ground into powder using liquid nitrogen, and proteins were extracted following previously established methods [25]. The protein extract was then centrifuged to obtain the protein sample, which was stored at −80 °C.

We added 0.5 M of TEAB to 100 μg of protein sample to dilute it and reduce the urea concentration to less than 2 M. Trypsin was added to the sample at a ratio of 1:20 (enzyme-to-protein), followed by mixing and centrifugation. The resulting mixture was incubated at 37 °C for 4 h, then desalted and dried. The peptides were labeled with tandem mass tag (TMT) reagents (BGI Genomics Co., Ltd., Shen Zhen, China) per the reagent guidelines. The peptides were subsequently mixed, desalted, and dried.

Subsequently, the labeled peptide samples were subjected to liquid chromatography using a Shimadzu LC-20AB system. The dried peptide samples were then separated by elution with mobile phase A and mobile phase B. The components of the two mobile phases were 5% acetonitrile and 95% acetonitrile, respectively, and the pH was 9.8. The components were collected at a wavelength of 214 nm and freeze-dried.

The peptides were suspended using mobile phase A, and the resulting solution was centrifuged at 2000× *g* for 10 min. The supernatant obtained after centrifugation was separated using an Easy-n LC 1200 system (Thermo Fisher Scientific, San Jose, CA, USA). The sample was loaded into a C1 self-packed column with mobile phase B in the column, and the liquid chromatography system was linked to the mass spectrometer.

Following separation via liquid chromatography, the peptide was ionized via nano-electrospray ionization (nano-ESI) and detected using the Orbitrap Exploris 480 tandem mass spectrometry in DDA mode (Thermo Fisher Scientific, San Jose, CA, USA). The parameters of the instrument were as follows: the ion source voltage was set to 2.1 V, while M1 and M2 had mass ranges of 350–1600 *m*/*z* and 100 *m*/*z*, respectively. The M1 and M2 also had resolutions of 60,000 and 15,000, respectively. The selection criteria for ion fragmentation in MS2 were based on an ion charge of 2+ to 7+ and a first parent ion peak intensity greater than 50,000. The ion fragmentation method was set to HCD, and broken ions were detected using Q-Exactive Orbitrap. Additionally, dynamic exclusion was set to 30 s, and the AGC values were set to MS1 1E6 and MS2 1E5.

### 2.4. Expression and Function Analysis

The software program IQuant [26] was used to analyze labeled peptides with isobaric tags and identify confident proteins based the “simple principle” (the parsimony principle). The protein false discovery rate (FDR) of 1% was estimated at the protein level to control the rate of false positive results [27]. If the fold change was greater than 1.2 and *p*-value less than 0.05, then the protein would be considered significantly upregulated. Conversely, if the fold change was less than 0.833333 and *p*-value less than 0.05, then the protein would be considered significantly downregulated.

The software program Proteome Discoverer (https://thermo-proteome-discoverer.software.informer.com/, accessed on 20 February 2022) was used to search MS/MS spectra against a database of transcriptome-based protein sequences [17] for the identification of parent proteins. The highest score for a specific peptide mass was considered the best match to that predicted in the database. Parameters for protein searching were set by considering tryptic digestion with two missed cleavages, carbamidomethylation of cysteines as fixed modification, and oxidation of methionines and protein N-terminal acetylation as variable modifications. Peptide spectral matches were validated based on q-values at a 1% false discovery rate (FDR), which indicates the probability of identifying false positives in the dataset.

Next, the identified protein IDs were converted to UniProt IDs and mapped to GO IDs using the software program InterProScan (v.5.14-53.0, http://www.ebi.ac.uk/interpro/, accessed on 21 February 2022) for annotation of protein function. KAAS (v.2.0 http://www.genome.jp/kaas-bin/kaas_main, accessed on 21 February 2022) was used to identify KEGG pathways significantly enriched in the differentially expressed proteins to explore the pathways implicated in the response to heat stress. In addition, functional enrichment analysis including GO and KEGG were performed using Goatools (https://github.com/tanghaibao/goatools, accessed on 21 February 2022) and KOBAS (http://kobas.cbi.pku.edu.cn, accessed on 21 February 2022) to identify DEGs which were significantly enriched in biological pathways at a Bonferroni-corrected *p*-value ≤ 0.05 compared with the whole-transcriptome background. The TMT-based proteomics datasets have been deposited in ProteomeXchange with identifier PXD035618.

### 2.5. Quantitative Real-Time PCR

A total of five genes were selected for expression quantification analysis by using quantitative real-time PCR (qPCR) to verify TMT-based proteomics results. For each sample, 1–2 µg total RNA was used in cDNA synthesis using GoScript^TM^ Reverse Transcription System (Promega, Clontech, Madison, WI, USA). Specific primers were designed (Table A1) based on mRNA sequences and synthesized by Sangon Biotech Co., Ltd, Shanghai, China. qPCR was performed using Go Taq^®^ qPCR and qPCR Systems (Promega, Clontech, Madison, WI, USA). The reaction was carried out in a total volume of 20 µL.

### 2.6. Statistic Analysis

In order to evaluate the statistical significance of the differences observed between control and heat treatment groups, analysis of variance was carried out using the software program IBM SPSS 22.0 at a significance level of 0.05. All data presented are means with standard deviation calculated from the triplicated samples from each group.

## 3. Results

### 3.1. Activity of Cellular Enzyme

The results of the study indicated that the activities of SOD (Figure 1A) and ACP (Figure 1B) were significantly higher in the two high-temperature treatment groups compared to the control group. Additionally, the LDH activity in the 26 °C treatment group was significantly higher than that in the control group, but the opposite was observed in the 33 °C treatment group (Figure 1C). Moreover, the cellular content of LZM was significantly increased in the two treatment groups compared to that of the control group (Figure 1D). These results suggest that the antioxidant and immune systems respond to heat stress. Additionally, the opposite responses of LDH activity in the two high-temperature treatment groups suggest that anaerobic respiration may be affected by different temperature conditions.

### 3.2. Differential Protein Expression Profiles

In total, 1,101,958 spectrums were generated, and 67,221 spectrums found matches in the database (Table A2). In all, 7559 mapped proteins from our proteomic analysis, namely collagen type VI alpha, glucuronosyltransferase, protein transport protein SEC61 subunit gamma and adenylyltransferase, were highly expressed proteins under normal conditions. Compared to the control group, the high-temperature conditions resulted in the identification of 1652 to 1878 differentially expressed proteins (DEPs), which accounted for 21.8% to 24.8% of all the detected proteins (Table A3). The expression pattern and number of differentially expressed proteins (DEPs) were similar in both the 26 °C and 33 °C treated groups. Compared with the control group, a total of 897 DEPs were identified and shared between the two groups (Figure 2A,B). Specifically, the 26 °C treated group had 763 significantly upregulated DEPs and 889 significantly downregulated DEPs, while the 33 °C treated group had 1051 significantly upregulated DEPs and 827 significantly downregulated DEPs (Figure 2C,D). Of these, the DEPs in both the 26 °C and 33 °C treated groups were enriched in similar KEGG pathways, including protein digestion and absorption, p53 signaling pathway, longevity regulating pathway, lysosome, fatty acid metabolism, DNA replication, fat digestion and absorption, and apoptosis (Figure 3A,B).

The differentially expressed proteins (DEPs) in the 26 °C treatment group were enriched in the gene ontology (GO) categories. In terms of molecular function (MF), the majority of proteins were involved in “binding”, followed by those related to “catalytic activity”. In terms of cellular component (CC), the largest group of proteins was involved in the “membrane”. With respect to biological process (BP), the majority of proteins participated in “metabolic processes” (Figure 3C). Furthermore, the expression patterns of DEPs in the 26 °C treatment group were similar to those in the 33 °C treatment group (Figure 3D).

### 3.3. Nutrients Digestion and Absorption

High-temperature treatment significantly changed the expression of 28 proteins, and the expression patterns of these DEPs were similar in both heat-treatment groups (Figure 4). Among these DEPs, 9 were involved in protein digestion and absorption, 5 were involved in fat digestion and absorption, and 3 were involved in carbohydrate digestion and absorption. In addition, 6 proteins secreted by the pancreas, including muscarinic acetylcholine receptor M3 (CHRM3), cholecystokinin A receptor (CCKAR), carbonic anhydrase 2 (CA2), secretory phospholipase A2 (PLA2G), Ras-related C3 botulinum toxin substrate 1 (RAC1), and Ras-related protein Rab-3D (RAB3D), were significantly changed under heat conditions.

### 3.4. Specific Regulation of Metabolism Pathways

After treatment with high temperature, Acetyl-CoA synthetase (ACS) was upregulated in both the 26 °C and 33 °C treatment groups. Moreover, several enzymes in the citric acid (TCA) pathway, such as 2-oxoglutarate dehydrogenase E1 component (OGDH), malate dehydrogenase (MDH), and succinate dehydrogenase (SDH), were similarly regulated in response to acclimation in both the 26 °C and 33 °C groups. Notably, no enzyme in the TCA pathway was downregulated under either treatment (Figure 4). In the fatty acid degradation pathway, CPT-1, ACO, and EHHADH were upregulated in both the 26 °C and 33 °C treatment groups (Figure 4).

Overall, several enzymes involved in amino acid metabolism were differentially regulated in response to the two heat treatments. ACAA, which catalyzes acetyl-CoA production by isoleucine degradation, was upregulated in the heat treatments, while ACAT, which catalyzes the pathway from leucine to acetyl-CoA, was downregulated in the heat treatment groups (Figure 4). ALT was increased in the heat treatment groups along the pathway from alanine, aspartate, and glutamate metabolism to produce fumarate, which are substrates for reactions in the TCA cycle (Figure 4). These findings suggest that energy metabolism-related pathway proteins respond to heat stress in *M. coruscus*.

### 3.5. Signaling Pathway of Stress Response to Heat Stress

The proteomic data also showed significant regulation of proteins associated with environmental stress. When compared to the control, caspase 3 proteins were significantly upregulated in the p53 signaling pathway in the 26 °C and was significantly downregulated in the 33 °C (Figure 4). In the MAPK signaling pathway, p38 showed no significant change in the 33 °C treatment group, but was downregulated in the 26 °C treatment group (Figure 4). Additionally, the apoptosis-related proteins, such as CHEK2, Bcl-xL, and caspase family proteins, were significantly changed after treatment with high temperature.

### 3.6. Validation of Proteomic Results Using Quantitative Real-Time PCR

To confirm the expression patterns of differentially expressed proteins (DEPs) following heat stress in *M. coruscus*, changes in transcript levels were evaluated. Five heat shock protein (HSP) genes, including HSP1A, HSP16.1, HSPBP1, HSP110, and CRYAB, were selected, and their expression profiles were measured via qPCR analysis. The mRNA expression trends of all examined genes were consistent with the protein levels observed in the proteomic analysis (Figure 5).

## 4. Discussion

The normal life activities of mussels are vulnerable to external environmental stressors, including hypoxia, sea warming, and acidification [28,29,30]. Exposure to high-temperature conditions significantly downregulates the organism’s energy reserves and weakens its immune function [31,32]. As the molecular response mechanism related to injury and death in mussels after high-temperature stress remains unclear, we examined the responses of *M. coruscus* to high-temperature exposure to investigate the molecular and cellular mechanisms involved. This study found that high temperatures regulate the antioxidant system, immune system, and anaerobic respiration of *M. coruscus*. Additionally, proteomic analysis showed that fatty acid metabolism, amino acid metabolism, the p53 signaling pathway, and the MAPK signaling pathway are related to the high-temperature response of *M. coruscus* (Figure 6).

### 4.1. Antioxidant and Immune Function under High Temperature

Sharp changes in water temperature can have significant effects on aquatic organisms, with excessive reactive oxygen species (ROS) and free radicals being the most common phenomena in heat stress responses [33]. Thermal stress directly impacts the metabolism of organisms, leading to metabolic disorders and ROS accumulation [34,35]. In this study, the activity of the antioxidant enzyme SOD was significantly increased after heat stress in two heat-treated groups, indicating that the enzyme responds to ROS production during heat stress in the digestive gland tissues of *M. coruscus*. Similar observations of increased antioxidant enzymatic activities have been reported in mussels after thermal exposure [36,37]. The effective induction of antioxidant enzymes in tissues may contribute to clearing accumulated peroxides under heat stress. For species with weak antioxidant capacity, the activities of SOD will gradually reduce with an increase in the number of free radicals, resulting in a greater degree of oxidative damage to the body [38]. In this study, SOD activities increased gradually with the increase of temperature, indicating that *M. coruscus* responds to heat stress by increasing its antioxidant capacity.

Ambient stress, such as heat shock, can decrease the oxygen concentration in organisms [39]. As a result of oxygen limitation, animals typically employ anaerobic glycolysis to meet their energy demand [40,41]. This upregulation of anaerobic pathways is typically indicated by an increase in the activity of the metabolic enzyme LDH [42]. However, the present study indicates that the activity of LDH decreased in *M. coruscus* when exposed to extreme heat, suggesting a decrease in anaerobic glycolysis function. This could be a result of *M. coruscus* responding to the extreme temperature conditions.

ACP is an important phosphatase in marine organisms, participating in degradation of foreign protein, carbohydrates, and lipids [43]; taking part in dissolution of dead cells; and acting as an ideal stress indicator in biological system [44]. In this study, the activities of ACP increased significantly in the digestive gland, which may indicate that *M. coruscus* responds to heat stress via the hydrolysis of high-energy phosphate bonds to liberate phosphate ions to combat stressful conditions or high metabolic rates [45].

In this study, the subjects’ content of LZM was found to have significantly increased. This could be a result of the activation of the immune system, which is an additional stress response. The present study showed that LZM significantly increased at both 26 °C and 33 °C. More relevant literature further showed that there was a tendency for the content of LZM to decrease with increasing temperature or increasing treatment time, with a trend of increasing and then decreasing [46,47].

### 4.2. Effects of High Temperature on Nutrient Digestion

Bivalves’ digestive gland, also called the hepatopancreas, is comprised of digestive cells [48]. A recent analysis of physiological indicators revealed that the digestive enzyme activities of *M. coruscus* are affected by temperature [15]. In the present study, we observed that the biological pathways related to nutrient digestion and absorption proteins were significantly changed under heat conditions. Notably, most of the proteins involved in digestion were found to be downregulated under heat stress. Previous studies have indicated that the activity of amylase, maltase, trypsin, and pancreatic lipase can be enhanced in response to a rise in temperature [49,50,51]. These results suggest that the digestive function of *M. coruscus*’ hepatopancreas may be weakened under heat stress.

### 4.3. Effects of High Temperature on Signaling Pathway of Stress

Heat stress can activate multiple signaling pathways and downstream responses to cope with the effects of heat stress on the body. In this study, we observed that proteins involved in the p53 signaling pathway, MAPK signaling pathway, and apoptosis pathway were significantly changed in *M. coruscus* exposed to high temperatures. The p53 pathway, which is critical for cell apoptosis, was significantly changed, indicating that high temperatures might harm *M. coruscus* cells. Additionally, the MAPK signaling pathway was also significantly changed. Mitogen-activated protein kinase (MAPK) is a group of conserved protein kinases that can transmit extracellular signals into the cell, resulting in various cellular responses such as cell proliferation, differentiation, and apoptosis [52]. The MAPK family includes the ERK, JNK, and p38MAPK signaling pathways. We observed a reduction in p38 in the p38MAPK signaling pathway in response to temperature stimulation under heat stress. This is consistent with previous studies that have shown that reducing p38 can eliminate damaged cells in response to stimuli [53,54]. Overall, *M. coruscus* may respond to heat stress by regulating apoptosis.

### 4.4. Effects of High Temperatures on Metabolism

The primary challenge for an organism under harsh environmental conditions is to optimize its energy requirements to maintain basic physiological processes. Acetyl-CoA synthetase was found to be upregulated in *M. coruscus* at 26 °C and 33 °C. In high-temperature environments, *M. coruscus* may activate the TCA cycle to increase ATP production by upregulating Acetyl-CoA synthetase. Furthermore, we observed that key enzymes involved in fatty acid metabolism and isoleucine metabolism were upregulated after exposure to high temperatures, indicating that these pathways may aid in adaptation to heat stress. Through these modifications, the supplementation of acetyl-CoA from D-glyceraldehyde-3P and long-chain lipids were enhanced [55].

It is noteworthy that many proteins involved in the TCA cycle were upregulated in the heat-treated group, indicating that the energy demand for *M. coruscus*’s metabolism increased with increasing temperature within a certain temperature range.

*M. coruscus* may respond to environmental stress by upregulating protein factors in the TCA cycle [56], while the enzyme activity in glycolytic and TCA cycle may be increased to enhance adaptation to environmental stress [57]. Glycolysis-related protein factors have been found to increase significantly under air exposure stress in the Crassostrea gigas [23]. When encountering variable temperatures, fatty acid metabolism provides advantages to facilitate variable resistance. In this study, the key proteins of amino acid metabolism in *M. coruscus* were upregulated, and the results suggest that enhanced fatty acid metabolism is conducive to adaptation to heat stress. We also observed that alanine production pathways in amino acid metabolism were upregulated, and *M. coruscus* may enhance environmental adaptation by entering the TCA cycle through alanine.

## 5. Conclusions

To investigate the response mechanism of *M. coruscus* to heat stress, we conducted an analysis of enzyme activity and TMT quantification in the digestive gland tissue of *M. coruscus* after exposure to high temperatures. Changes in four enzymes showed that *M. coruscus* can respond to damage caused by high-temperature stress through the antioxidant system, immune system, and anaerobic respiration. Proteomic analysis and HSP expression levels showed that *M. coruscus* can maintain normal cell morphology and function through the process of apoptosis mediated by p53 and MAPK signaling pathways. Additionally, *M. coruscus* may use fat and amino acids to meet energy requirements under high-temperature stress via the TCA cycle pathway. These results will provide a useful reference for further understanding response mechanism to heat stress in marine invertebrates such as mollusks.

## Figures and Tables

**Figure 1 animals-13-02248-f001:**
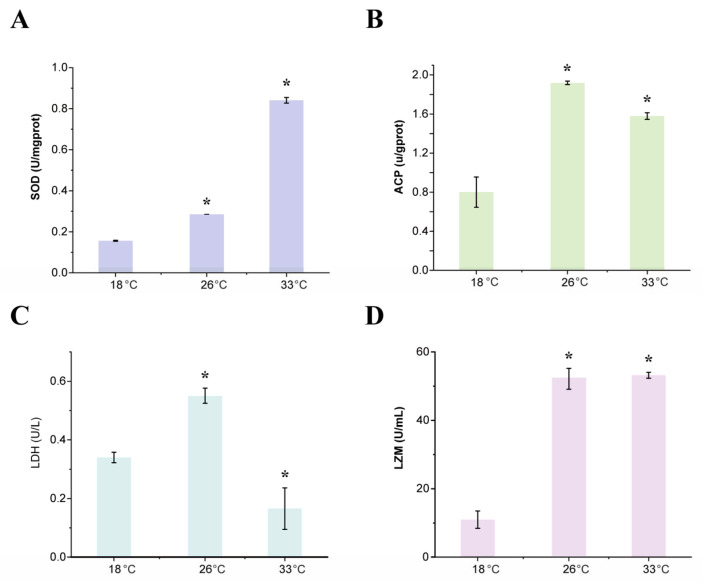
Activity of SOD (U/mgprot, (**A**)), ACP (u/gprot, (**B**)), LDH (U/L, (**C**)) and LZM (U/mL, (**D**)) in *M. coruscus* exposed to heat stress. * indicates significant differences.

**Figure 2 animals-13-02248-f002:**
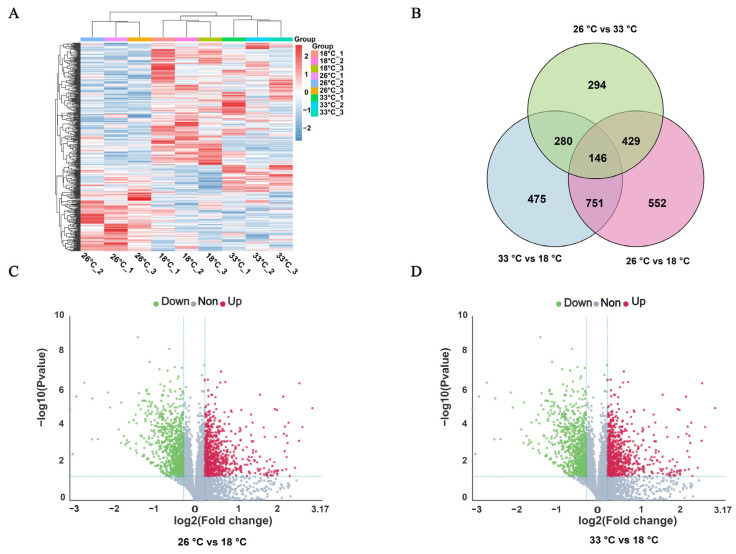
Proteomics profiles of *M. coruscus* under three different temperature conditions. (**A**) A heatmap was used to classify the DEP expression patterns in all groups; the *X*-axis represents the individuals in different temperature groups. (**B**) Venn diagram of DEPs in 26 °C vs. 18 °C, 33 °C vs. 18 °C, and 26 °C vs. 33 °C groups. (**C**) Volcano plot showing the up- and downregulated DEPs in 26 °C vs. 18 °C groups. (**D**) Volcano plot showing the up- and downregulated DEPs in 33 °C vs. 18 °C groups.

**Figure 3 animals-13-02248-f003:**
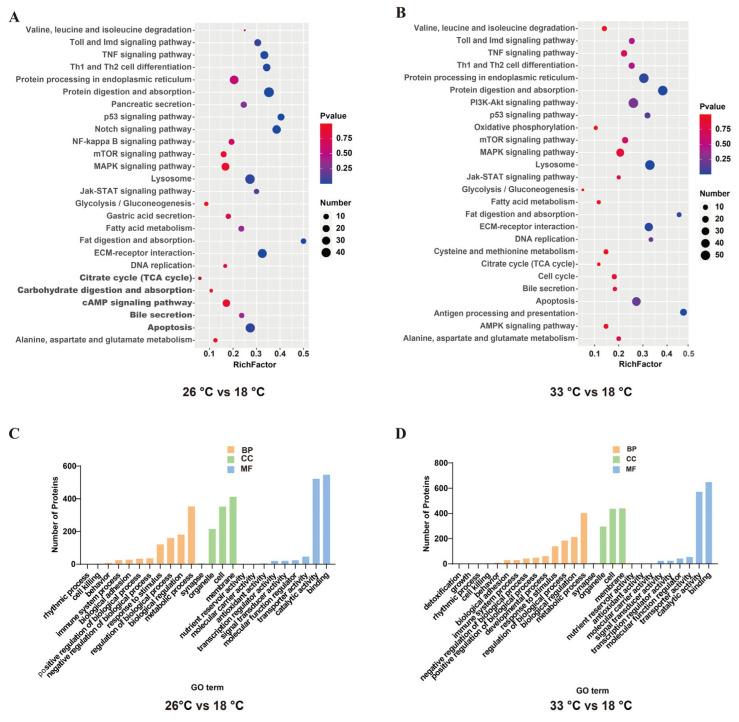
Functional enrichment analysis identified major biological processes and pathways after infection. KEGG enrichment analysis of regulated proteins at 26 °C vs. 18 °C groups (**A**) and 33 °C vs. 18 °C groups (**B**). GO enrichment analysis of regulated proteins at 26 °C vs. 18 °C groups (**C**) and 33 °C vs. 18 °C groups (**D**).

**Figure 4 animals-13-02248-f004:**
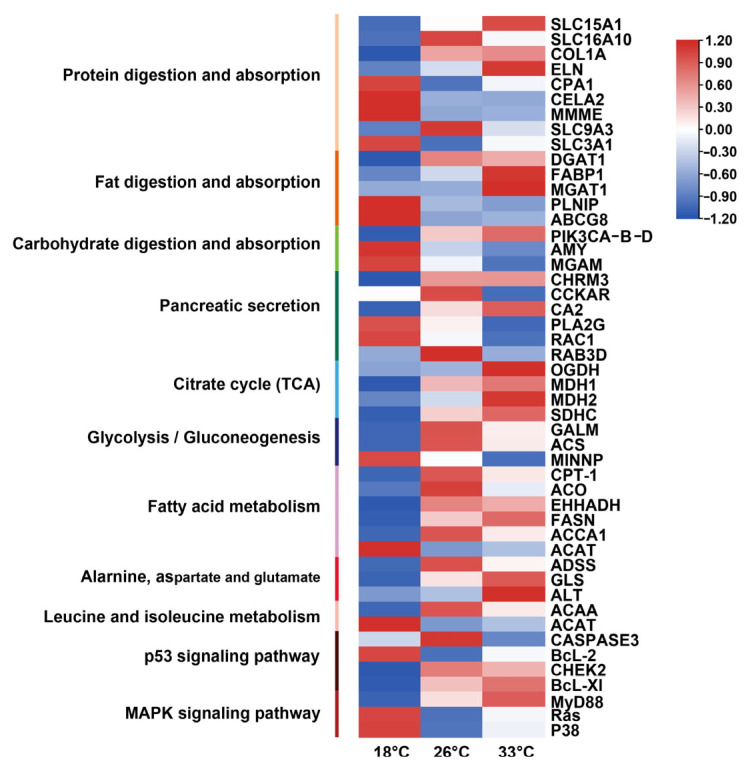
Biological processes enriched by DEPs after treatment with heat stress. The heatmap color strength represents homogenized protein expression, from blue (lowest) to red (highest).

**Figure 5 animals-13-02248-f005:**
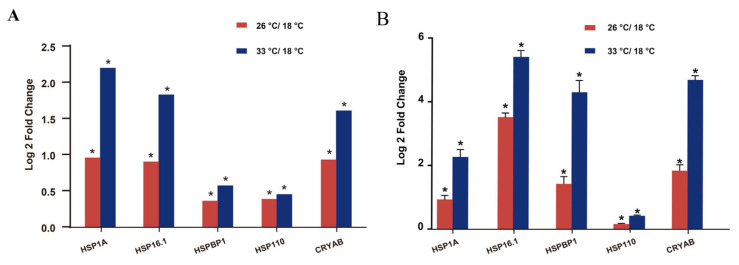
Expression fold changes of selected genes under heat stress. (**A**) Five differently expressed proteins from the HSP family including HSP1A, HSP16.1, HSPBP1, HSP110, and CRYAB. (**B**) qPCR analysis of five genes involved in HSP, including HSP1A, HSP16.1, HSPBP1, HSP110 and CRYAB. *β*-actin was set as reference gene, analysis of significance was carried out using *t*-test at the significance level of 0.05; * indicates significant differences.

**Figure 6 animals-13-02248-f006:**
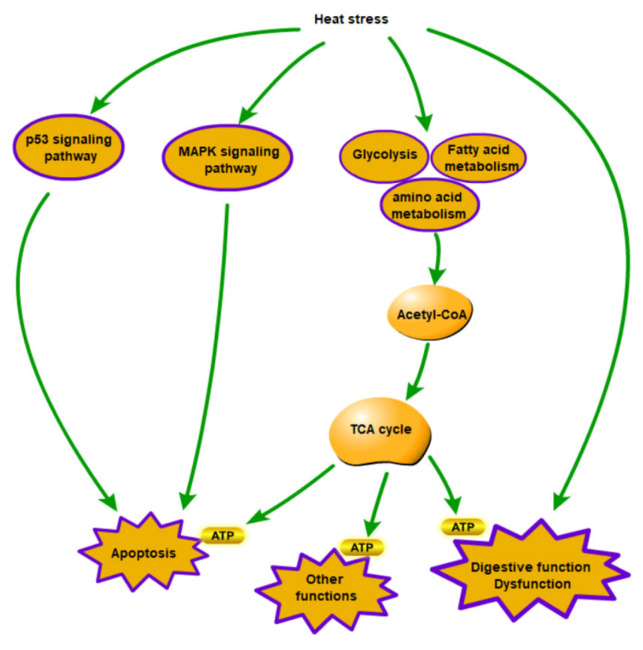
Schematic representation of biological pathways in *M. coruscus* under heat stress.

## Data Availability

The datasets supporting the results of this article are included within the manuscript and Appendix A.

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
