# Peer review of "Insights into the Response in Digestive Gland of Mytilus coruscus under Heat Stress Using TMT-Based Proteomics"

_animals, 2023, doi:10.3390/ani13142248_

Round 1

Reviewer 1 Report

This article combines M. coruscus and proteomics to provide insights into how the digestive glands of mussels respond and adapt to high temperatures. The article systematically describes the protein expression in the digestive glands of M. coruscus at high temperatures in terms of apoptosis, immunity, amino acid metabolism and lipid metabolism, which is of high value for farm M. coruscus culture and for investigating the adaptation of M. coruscus to environmental temperature changes.

1.     The choice of biological enzymes is better supported by the literature.

2.     The detoxification and growth of the GO enrichment plot in Figure 3D is marked in red, but it is not specifically mentioned in the article, so I don't know what it means, so we suggest removing the red font.

3.     The diagram in Fig. 4 is not labelled A & B and is omitted, but the diagram notes and the article have A & B.

4.     In line 213, Acetyl-CoA synthetase (ACS) is not shown in Figure 4A and B in English abbreviations, where is this ACS seen to be upregulated at 26°C and 33°C?

5.     Lines 237-238, the article mentions that "Caspase 3 protein was significantly upregulated in the p53 signalling pathway in the 26°C and 33°C treated groups compared to the control group", but the Caspase 3 protein in Figure 4B looks like it is downregulated at 33°C compared to the control group, please double check if the description is correct.

6.     Lines 241-244, please double check if the description is correct, apoptosis related proteins CHEK2 and BcL-xL are overall upregulated outside of BCL-2 which is downregulated and Caspase 3 protein is like downregulated at 33°C compared to the control. So the article describes, "But apoptosis-related proteins such as caspase family proteins were upregulated overall." Please double check if the description is correct.

7.     The comparison between the qRT-PCR results and the proteomics data in Figure 5 is not obvious, and it is not possible to visualize whether the proteomics data are credible and usable. It is recommended that all of them be replaced by bar charts with the expression trends of the five genes in the proteomics data and the qRT-PCR results.

9.Lines 163-163 say that LZM was significantly increased in both experimental groups, and by lines 295-297 it becomes: "There was a tendency for the LZM content to decrease with increasing temperature or longer treatment time, with a tendency to increase and then decrease". This paragraph is still quoted from the literature and it is not clear what exactly it is trying to say.

I think it could be changed to read, "The present study showed that LZM was significantly increased at both 26°C and 33°C. More relevant literature further showed that there was a tendency for the content of LZM to decrease with increasing temperature or increasing treatment time, with a trend of increasing and then decreasing."

10.In line 321, the word "moderate" appears for the first time in the text, suggesting that this word be followed by the temperature corresponding to the experiment in this text, whether 26°C or 33°C.

11.In lines 321-322, "This is consistent with other studies" suggest adding support from the literature.

12.In 4.4, the article once again suggests that Acetyl-CoA synthetase expression is upregulated, but looking around the entire article, none of the graphs mention Acetyl-CoA synthetase or its abbreviation ACS expression, please double check.

13. In line 329-331, "On the other hand, we founded that 329 the key enzymes involved in fatty acid metabolism, leucine and isoleucine metabolism 330 were up-regulated after treating with high temperature", the description of "leucine" in line 224 of the article is significantly down-regulated, not up-regulated as stated in this sentence. Please double check.

14. Line 346, "We also found that pyruvate production pathways in amino acid metabolism were upregulated" where the word "pyruvate" is mentioned for the first time. pyruvate" is mentioned for the first time, but only line 225 has the word "alanine" associated with it, and there is no mention of "pyruvate" in the legend. In line 347 it becomes "alanine" again. Please check again whether it is "pyruvate" or "alanine".

15. The conclusion could be a little more about the next research directions, progress, and what the findings of this article have taught us.

16. Should the reference article format be abbreviated last name or first name.

17. Species names in the text should be in italics, including references

Author Response

  1. The choice of biological enzymes is better supported by the literature.

Response: The activities variations of these enzyme have been reported in M. coruscus treated with environmental stress in references including ‘Combined toxic effects of nanoplastics and norfloxacin on mussel: Leveraging biochemical parameters and gut microbiota’ (doi:10.1016/j.scitotenv.2023.163304) and ‘Microplastics aggravate the adverse effects of BDE-47 on physiological and defense performance in mussels’ (doi:10.1016/j.jhazmat.2020.122909). And the above references have been added in revised MS.

  1. The detoxification and growth of the GO enrichment plot in Figure 3D is marked in red, but it is not specifically mentioned in the article, so I don't know what it means, so we suggest removing the red font.

Response: The mark with red font has removed in the revised Fig. 3D.

  1. The diagram in Fig. 4 is not labelled A & B and is omitted, but the diagram notes and the article have A & B.

Response: The figure 4 has been modified and the part of A and B was merged in revised MS.

  1. In line 213, Acetyl-CoA synthetase (ACS) is not shown in Figure 4A and B in English abbreviations, where is this ACS seen to be upregulated at 26°C and 33°C?

Response: The ACS has been added to revised Figure 4, and the ACS was upregulated at both 26°C and 33°C.

  1. Lines 237-238, the article mentions that "Caspase 3 protein was significantly upregulated in the p53 signalling pathway in the 26°C and 33°C treated groups compared to the control group", but the Caspase 3 protein in Figure 4B looks like it is downregulated at 33°C compared to the control group, please double check if the description is correct.

Response: The sentence of "Caspase 3 protein was significantly upregulated in the p53 signalling pathway in the 26°C and 33°C treated groups compared to the control group" has been replaced by "caspase 3 proteins was significantly upregulated in the p53 signaling pathway in the 26°C and was significantly downregulated in the 33°C"

  1. Lines 241-244, please double check if the description is correct, apoptosis related proteins CHEK2 and BcL-xL are overall upregulated outside of BCL-2 which is downregulated and Caspase 3 protein is like downregulated at 33°C compared to the control. So the article describes, "But apoptosis-related proteins such as caspase family proteins were upregulated overall." Please double check if the description is correct.

Response: The sentence was replaced by “Additionally, the apoptosis-related proteins, such as CHEK2, Bcl-xL, and caspase fami-ly proteins, were significantly regulated after treatment with high temperature.”

  1. The comparison between the qRT-PCR results and the proteomics data in Figure 5 is not obvious, and it is not possible to visualize whether the proteomics data are credible and usable. It is recommended that all of them be replaced by bar charts with the expression trends of the five genes in the proteomics data and the qRT-PCR results.

Response: The fig.5 has been modified and the expression trends of these five genes in fig.5 have we shown by bar charts.

  1. Lines 163-163 say that LZM was significantly increased in both experimental groups, and by lines 295-297 it becomes: "There was a tendency for the LZM content to decrease with increasing temperature or longer treatment time, with a tendency to increase and then decrease". This paragraph is still quoted from the literature and it is not clear what exactly it is trying to say.

I think it could be changed to read, "The present study showed that LZM was significantly increased at both 26°C and 33°C. More relevant literature further showed that there was a tendency for the content of LZM to decrease with increasing temperature or increasing treatment time, with a trend of increasing and then decreasing."

Response: Thank you for your advice. These sentences were replaced by "The present study showed that LZM was significantly increased at both 26°C and 33°C. More relevant literature further showed that there was a tendency for the content of LZM to decrease with increasing temperature or increasing treatment time, with a trend of increasing and then decreasing."

  1. In line 321, the word "moderate" appears for the first time in the text, suggesting that this word be followed by the temperature corresponding to the experiment in this text, whether 26°C or 33°C.

Response: The word of "moderate" means 26°C or 33°C, and it has been deleted in revised MS to avoid misunderstanding.

  1. In lines 321-322, "This is consistent with other studies" suggest adding support from the literature.

Response: These referable articles, including “DOI:10.26355/eurrev_201902_17029” and “DOI: 10.1016/j.brainres.2020.146932” have been added in lines 321-322, and listed in references in revised MS.

  1. In 4.4, the article once again suggests that Acetyl-CoA synthetase expression is upregulated, but looking around the entire article, none of the graphs mention Acetyl-CoA synthetase or its abbreviation ACS expression, please double check.

Response: The ACS has been added to Figure 4.

  1. In line 329-331, "On the other hand, we founded that 329 the key enzymes involved in fatty acid metabolism, leucine and isoleucine metabolism 330 were up-regulated after treating with high temperature", the description of "leucine" in line 224 of the article is significantly down-regulated, not up-regulated as stated in this sentence. Please double check.

Response: The word of " leucine " has been deleted.

  1. Line 346, "We also found that pyruvate production pathways in amino acid metabolism were upregulated" where the word "pyruvate" is mentioned for the first time. pyruvate" is mentioned for the first time, but only line 225 has the word "alanine" associated with it, and there is no mention of "pyruvate" in the legend. In line 347 it becomes "alanine" again. Please check again whether it is "pyruvate" or "alanine".

Response: The word of "pyruvate" has been corrected by "alanine" in revised MS.

  1. The conclusion could be a little more about the next research directions, progress, and what the findings of this article have taught us.

Response: Thank you for your advice. The sentence of “These results will provide a useful reference for further understanding response mechanism to heat stress in marine invertebrates like molluscs.” was added in revised MS.

  1. Should the reference article format be abbreviated last name or first name.

Response: The reference article format is abbreviated first name, and orders of name in all references were corrected in revised MS.

  1. Species names in the text should be in italics, including references

Response: It was corrected in all text.

Reviewer 2 Report

Major concern:

1.      The abstract should be reorganized to make the experimental design and results clearer.

2.      The introduction is poorly organized and does not make sense in its current form. The authors should summarize what we have already known and what remain unclear. Based on these, the authors should put forward their own hypothesis or scientific questions. Without these, the study and its findings may hardly be integrated into our knowledge system.

3.      Experimental design and analyses. Detailed explanation for the temperature selection is required, e.g., whether the temperatures used to induce heat stress are environmental relevant or theoretically significant for the studied animal? The comparisons between the two treatment groups are also important. For example, the authors may consider to present the changes caused by 26 and 33 °C in the same panel (Figure 3 A and B). The author should explain why these two heat stresses caused similar or different outcomes with the consideration of the physiological significance of these two temperatures. For omics, multivariate analysis (e.g., PCA and PLS) is quite important, as it well present the similarity between samples, as well as the variation trends of different treatment groups. The authors should take it into consideration.

4.      The authors have not well integrated their findings, especially the relativeness between changes in different biological processes. This is important for omics, which generate large amount of differently expressed genes/proteins/metabolites. The authors may consider to summarize their findings with a sketch map.

5.      English editing by native speakers or chatGPT is required.

Line 19 Avoid abbreviation in abstract.

Line 22 ‘replace’ was with ‘were’. Moreover, rephrase this sentence. It seems to be better to exchange the place between LZM and LDH, to put the enzymes together.

Line 22 What does ‘significantly regulated’ mean?

Line 22-23 It should be 1,652 and 1,878. Similar errors should be checked throughout the manuscript.

Line 23 How did your DEP screened? The DEP between 26 and 33 °C or other comparisons?

Line 38-40 References are required.

Line 40-41 grammatical mistake.

Line 44 provide the full name of this species when it appears for the first time in the main text.

Line 46 Mytilus should be in italic, and check this throughout the manuscript.

Line 46-48 Rephrase this sentence, e.g., compared to another aquatic shellfish M. galloprovincialis. Note that as the two Mytilus species share the same genus name, so you can use the abbreviation M. galloprovincialis for Mytilus galloprovincialis when Mytilus sp. has already appeared in the main text.

Line 69 the temperatures. ‘A, B, and C’ differs with ‘A, B and C’ in that the B and C are separated groups. Check this throughout your manuscript.

Line 77 insert ‘and’ before LDH

Line 82 replace 3 to three. Note the use of numbers in scientific writings.

Line 134-136 Replace ‘Foldchange’ and ‘Pvalue’ to ‘fold change’ and ‘p value’, respectively, even these two terms are the parameter names in a certain software.

Line 135-136 The statistics. As there are large number of proteins, did you consider to adjust your p values (e.g., the false discovery rate or q values)?

Line 143 Please provide the URL.

Figure 2 ‘A vs B’ means the group B is used as a control, please check it whether you have used it correctly.

Figure 3 Increase the font size of the text. The caption should explain what does it mean for the items in red color (panel D)?

Figure 4 The figure does not include two panels.

Author Response

  1. The abstract should be reorganized to make the experimental design and results clearer.

Response: According to your suggestion, the abstract has been modified to “Ocean warming can cause injury and death in mussels, and is believed to be one of the main reasons for extensive die-offs of mussel populations worldwide. However, the biological processes by which mussels respond to heat stress are still unclear. In this study, we conducted an analysis of enzyme activity and TMT-labelled based proteomic in digestive gland tissue of Mytilus coruscus after exposure to high temperatures. Our results showed that the activities of superoxide dismutase, acid phosphatase, lactate dehydrogenase, and cellular content of lysozyme were significantly regulated in response to heat stress. Furthermore, many differentially expressed proteins involved in nutrient digestion and absorption, p53, MAPK, apoptosis, and energy metabolism were activated post-heat stress. These results suggest that M. coruscus can respond to heat stress through the antioxidant system, immune system, and anaerobic respiration. Additionally, M. coruscus may use fat, leucine, and isoleucine to meet energy requirements under high temperature stress via the TCA cycle pathway. These findings provide a useful reference for further exploration of the response mechanism to heat stress in marine molluscs.”.

  1. The introduction is poorly organized and does not make sense in its current form. The authors should summarize what we have already known and what remain unclear. Based on these, the authors should put forward their own hypothesis or scientific questions. Without these, the study and its findings may hardly be integrated into our knowledge system.

Response: According to your suggestion, the introduction has been modified in revised MS.

  1. Experimental design and analyses. Detailed explanation for the temperature selection is required, e.g., whether the temperatures used to induce heat stress are environmental relevant or theoretically significant for the studied animal? The comparisons between the two treatment groups are also important. For example, the authors may consider to present the changes caused by 26 and 33 °C in the same panel (Figure 3 A and B). The author should explain why these two heat stresses caused similar or different outcomes with the consideration of the physiological significance of these two temperatures. For omics, multivariate analysis (e.g., PCA and PLS) is quite important, as it well present the similarity between samples, as well as the variation trends of different treatment groups. The authors should take it into consideration.

Response: The temperature was set according to previous study, and the reference was added in the part of M&M in revised MS. And the detail has been described as ‘The researchers also pretested the temperature gradient (20, 25, 30, and 35 â—¦C) of this experiment, and the results showed that 24-26 â—¦C is the most suitable temperature for the growth of M. coruscus. The sea surface temperature changes in the intertidal zone of the East China Sea from March to May, which is between 10 and 18 â—¦C, and the experimental controls needed to be the same temperature as the bay. Therefore, 18 â—¦C was chosen as the control temperature (Additional Files 3). A temperature of 26 â—¦C was chosen because this is the highest temperature the category has experienced in the area, and 33 â—¦C was selected as the temperature condition for local temperature extremes.’ in the part of ref.17. There is no doubt that your suggestion is constructive about the comparison between the two temperatures. However, in this study, our aim is to obtain the holistic molecular regulatory mechanism under high temperature, and the similar or different outcomes between the two temperatures will be analyzed and verified in future experiments. In addition, according to your suggestion, the cluster analysis of the samples was carried out and shown in the revised Figure 2A to show the variation trends of all groups.

  1. The authors have not well integrated their findings, especially the relativeness between changes in different biological processes. This is important for omics, which generate large amount of differently expressed genes/proteins/metabolites. The authors may consider to summarize their findings with a sketch map.

Response: The summary content of major biological processes has been showed in added fig.6 in revised MS.

  1. English editing by native speakers or chatGPT is required.

Response: The MS has been edited by chatGPT.

  1. Line 19 Avoid abbreviation in abstract.

Response: The abbreviation has been deleted in abstract.

  1. Line 22 ‘replace’ was with ‘were’. Moreover, rephrase this sentence. It seems to be better to exchange the place between LZM and LDH, to put the enzymes together.

Response: The word of ‘was’ has been replaced by ‘were’. The place between LZM and LDH has been exchanged.

  1. Line 22 What does ‘significantly regulated’ mean?

Response: It means the contents of these enzymes were significantly changed under heat stress.

  1. Line 22-23 It should be 1,652 and 1,878. Similar errors should be checked throughout the manuscript.

Response: It has been corrected in revised MS.

  1. Line 23 How did your DEP screened? The DEP between 26 and 33 °C or other comparisons?

Response: The method of analysis of DEP was described in the section of materials and methods “When Foldchange > 1.2, Pvalue <0.05, the protein was considered to be significantly upregulated, and when Foldchange < 0.833333, Pvalue <0.05, protein was considered to be significantly down-regulated”. And the DEP was obtained in comparison of 26 °C with 18 °C and 33 °C with 18 °C, respectively.

  1. Line 38-40 References are required.

Response: The reference of “Chronic heat stress as a predisposing factor in summer mortality of mussels, Perna canaliculus” has been added.

  1. Line 40-41 grammatical mistake.

Response: The sentence has been modified to “In summer, affected by high temperature, mass mortality of oyster and the extensive infection of M. edulis with Marteilia pararefringens are common phenomenon.”

  1. Line 44 provide the full name of this species when it appears for the first time in the main text.

Response: ‘M. coruscus’ has been replaced by ‘Mytilus coruscus’.

  1. Line 46 Mytilus should be in italic, and check this throughout the manuscript.

Response: It has been corrected in all text.

  1. Line 46-48 Rephrase this sentence, e.g., compared to another aquatic shellfish M. galloprovincialis. Note that as the two Mytilus species share the same genus name, so you can use the abbreviation M. galloprovincialis for Mytilus galloprovincialis when Mytilus sp. has already appeared in the main text.

Response: The sentence of “In farmed Mytilus, compared with the other one aquatic shellfish Mytilus galloprovincialis, M. coruscus has the higher protein, crude fat, DHA, and trace element content, as well as a higher economic value.” Has been corrected to “In farmed Mytilus, compared to another aquatic shellfish M. galloprovincialis, M. coruscus has the higher protein, crude fat, DHA, and trace element content, as well as a higher economic value.” In addition, the word of “Mytilus edulis” was corrected to “M. edulis” in line 46 in revised MS.

  1. Line 69 the temperatures. ‘A, B, and C’ differs with ‘A, B and C’ in that the B and C are separated groups. Check this throughout your manuscript.

Response: “18 °C, 26 °C and 33 °C” has been replaced by “18 °C, 26 °C, and 33 °C” in all text.

  1. Line 77 insert ‘and’ before LDH

Response: The word of ‘and’ has been inserted before LDH.

  1. Line 82 replace 3 to three. Note the use of numbers in scientific writings.

Response: Thank you for your advice. ‘3’ has been replaced ‘Three’.

  1. Line 134-136 Replace ‘Foldchange’ and ‘Pvalue’ to ‘fold change’ and ‘p value’, respectively, even these two terms are the parameter names in a certain software.

Response: Thank you for your advice. ‘Foldchange’ and ‘Pvalue’ have been replaced ‘fold change’ and ‘p value’.

  1. Line 135-136 The statistics. As there are large number of proteins, did you consider to adjust your p values (e.g., the false discovery rate or q values)?

Response: Thank you for your advice. The threshold value of false discovery rate was selected according to our results, including the number of DEPs and the annotation of KEGG and GO, we think the p value used in this study was suitable based the above datasets.

  1. Line 143 Please provide the URL.

Response: It has been added in the revised MS.

https://proteomecentral.proteomexchange.org/cgi/GetDataset?ID=PXD035618

  1. Figure 2 ‘A vs B’ means the group B is used as a control, please check it whether you have used it correctly.

Response: It has been corrected in revised fig.2.

  1. Figure 3 Increase the font size of the text. The caption should explain what does it mean for the items in red color (panel D)?

Response: The font size has been increased and the red color has been deleted in revised fig.3.

  1. Figure 4 The figure does not include two panels.

Response: The two parts of A and B has been merged in revised Fig.4.

Reviewer 3 Report

The authors’ group has already published “Comparative transcriptomic analysis revealed changes in multiple signaling pathways involved in protein degradation in the digestive gland of Mytilus coruscus during high-temperatures” in CBP D (2023). I do not understand why this paper is not cited in the manuscript. It needs to be cited.

M & M section is too thin. It needs to be described in detail. Did the authors analyze the same mussles in this study as they used in the transcriptomic analysis?

What was derived from obtaining data for proteomic and transcriptomic analyses? It needs to be added to Discussion section.

Additionally, there are quite some odd wordings and use of grammar. I do suggest having some language editing done before submission.

I cannot recommend the acceptance of the present paper in this form for publication.

Author Response

  1. The authors’ group has already published “Comparative transcriptomic analysis revealed changes in multiple signaling pathways involved in protein degradation in the digestive gland of Mytilus coruscus during high-temperatures” in CBP D (2023). I do not understand why this paper is not cited in the manuscript. It needs to be cited.

Response: The reference has been added to revised MS in the part of Introduction and M&M and listed as ref.17.

  1. M & M section is too thin. It needs to be described in detail. Did the authors analyze the same mussles in this study as they used in the transcriptomic analysis?

Response: Some details was added in M&M in revised MS. The previous study of transcriptomic analysis and the proteomic analysis of this study were two completely separate experiments.

  1. What was derived from obtaining data for proteomic and transcriptomic analyses? It needs to be added to Discussion section.

Response: In this study, only the results of proteome analysis has been shown. Because the mussels we used in the transcriptomic analysis were not come from the same place as the mussels in this study, and the subtle differences was existent in the way of two experiments like treated time. Based on this, the analysis of the union of the two datasets has not been performed yet.

  1. Additionally, there are quite some odd wordings and use of grammar. I do suggest having some language editing done before submission.

Response: This manuscript has been edited in revised MS using the chatGPT tool.

  1. I cannot recommend the acceptance of the present paper in this form for publication.

Response: Thank you for your comments. According to all reviewers’ advices, the MS has been revised carefully. We look forward to receive your comments again.

Round 2

Reviewer 2 Report

The revised version is much better, but there are still some grammatical mistakes, e.g., line 241 and 249. What does you mean for 'significantly regulated'? The authors must improve their language and scientific writing.

Another major point. The statistics used in this study have not been fully described, e.g., line 258. Where is the error bar? How many replicates? Which statistical method? Besides, PCA or PCoA  is required to present the overall variation trends of the protein expression profile with the increase of the temperature.

  •  

Minor point

The subsections of 'Materials and Methods' should be set more properly. For example, the steps for the proteomics should be put together within the same subsection.

Figure 6 The author presented the associations between the enriched pathways, but they do not show the variation trends of these pathways.

I can't find the Ethical statement.

Author Response

  1. The revised version is much better, but there are still some grammatical mistakes, e.g., line 241 and 249. What does you mean for 'significantly regulated'? The authors must improve their language and scientific writing.

Response: The grammatical mistakes in the text have been corrected and highlighted in Line 241 and 249. 'significantly regulated' means significantly changed, and the word of 'significantly regulated' has been replaced by 'significantly changed' in revised MS.

  1. Another major point. The statistics used in this study have not been fully described, e.g., line 258. Where is the error bar? How many replicates? Which statistical method? Besides, PCA or PCoA is required to present the overall variation trends of the protein expression profile with the increase of the temperature.

Response: Error bars have been added in revised Figure 5. In this study, there individuals were used in qRT-PCR. The statistical method of t-test has been added in revised MS in line 261. The PCA analysis has been added in Appendix A as attached figure in revised MS.

Minor point

The subsections of 'Materials and Methods' should be set more properly. For example, the steps for the proteomics should be put together within the same subsection.

Response: The subheading of 2.4, 2.5 and 2.6 (these steps belonging to labelling and sequencing of proteins) have been deleted in revised MS.

Figure 6 The author presented the associations between the enriched pathways, but they do not show the variation trends of these pathways.

Response: These pathways were enriched by many differently expressed proteins, including up- and down-regulated DEPs. And we aim to find the affected biological process after treating with high temperature and to show the potential response mechanism in Mytilus coruscus. In addition, we will investigate the detail regulation patterns about these enriched pathways based on this study in next work.

I can't find the Ethical statement.

Response: Ethical statement is not application in invertebrates except cephalopods.

Reviewer 3 Report

The previous study on transcriptomic analysis of Mytilus coruscus by the authors’ group is interesting. I wished the authors would discuss the consistency and discrepancies between data from both transcriptomic and proteomic analyses. Though they cited the reference in Introduction, they do not seem willing to discuss them. I think this manuscript would be acceptable as a simple proteomics study.

Author Response

The previous study on transcriptomic analysis of Mytilus coruscus by the authors’ group is interesting. I wished the authors would discuss the consistency and discrepancies between data from both transcriptomic and proteomic analyses. Though they cited the reference in Introduction, they do not seem willing to discuss them. I think this manuscript would be acceptable as a simple proteomics study.

Response: Thank you for your suggestion and positive comment. Although some interesting results showed in transcriptomic analysis of Mytilus coruscus in our previous study, but we need more evident to reveal the relative biological pathways. In fact, many genes and pathways related studies are ongoing, and we hope to validate the results of previous high-throughput analyses with sufficient evidence in future. To help students (the main participants of the study) graduate on time, we have to publish our existing results in journals.